# OpenReview forum: "SECODEPLT: A Unified Benchmark for Evaluating the Security Risks and Capabilities of Code GenAI"
_NeurIPS.cc/2025/Datasets_and_Benchmarks_Track — NeurIPS 2025 Datasets and Benchmarks Track poster_

### Official Review · Reviewer_cur1 · 2025-06-25

**Rating:** 5
**Confidence:** 3

**Summary:**

This work presents SECCODEPLT, a multi-language benchmark for evaluating LLMs on security aspects of code generation. The dataset is constructed through a combination of human-curated seed collection and LLM-driven seed mutation to balance quality and scalability.  A comprehensive evaluation of LLM’s performance on code completion and instruction generation is conducted and results reveal substantial room for improvement.

**Additional Feedback:**

Duplicated reference 45 46

**Dataset Code Accessibility:**

Yes

**Dataset Code Comments:**

Dataset and code are public

**Ethical Considerations:**

No, there are no or only very minor ethics concerns

**Final Justification:**

The author's rebuttal and results addressed my concern, so I am raising the score to 5.

**Limitations Weaknesses:**

The test examples are generated from seeds, and this process may introduce duplication. Although the generated examples are deduplicated, it is unclear whether the LLM's poor performance is caused by the vulnerabilities and risks associated with the seeds, or the generated examples in the benchmark. For example, if a seed is directly presented to the LLM and the model fails to recognize its risk in code generation task, it is uncertain whether the LLM would also fail on all examples derived from this seed.

**Strengths Contributions:**

* This work proposes a new method for constructing benchmarks in a semi-automated manner. It combines manual seed selection and AI-assisted seed mutation to balance the quality and quantity of the generated benchmarks.
* Reasonable filtering and sanitization steps are taken during seed mutation process to ensure the fidelity of generated examples
* This benchmark covers multiple languages, and the approach is extensible and can be generalized to other languages and seed sources
* Authors conducted comprehensive evaluation on LLM, revealing that the sota LLMs perform poorly in security aspects on code generation tasks. These findings highlight directions and opportunities for future research

---

> ### Author Rebuttal · Authors · 2025-07-30
>
> We thank the reviewer for the insightful and positive feedback. Please see below for our response.
> ## 1. Mutation issues
>
> First, we would like to clarify that our mutation strategy does not only modify the function names and variable names. Our mutation strategies also include:
> - Loop Transformations: While to for loop; Do-while to while loop; Loop unrolling
> - Conditional Transformations: If-else to switch statements; Boolean expression simplification
> - Method Refactoring: Extract method refactoring; Inline method for simple functions
> - Code Style Changes: From procedural to functional style
>
> Here, we show an example with basic mutations (renaming) and Code Style Changes
> ```java
>     // other context...
>     public boolean isStringNotEmpty(String myString) throws Throwable {
>         boolean isGreaterThanZero = false;
>         if ((myString != null) & (myString.length() > 0)) {
>             IO.writeLine("The string length is greater than 0");
>             isGreaterThanZero = true;
>         }
>         return isGreaterThanZero;
>     }
> ```
>
> ```java
>     // other context...
>     public boolean validateString(String myString) throws Throwable {
>         return Optional.ofNullable(myString)
>             .filter(s -> s.length() > 0)
>             .map(s -> {
>                 try {
>                     IO.writeLine("The string length is greater than 0");
>                 } catch (Exception e) {}
>                 return true;
>             })
>             .orElse(false);
>     }
> ```
>
> Second, our mutation is mainly applied to Python, where we generate 9 mutations from one seed. This is because python does not have real-world codebases as seeds and we manually write the seeds. For C/C++ and Java, we obtain enough seeds from the real-world codebase and only generate 1–2 variants per seed. As such, the impact of the mutation is constrained in our benchmark.
>
> Third, we conducted a **similarity filter** to remove too similar data samples. Below, we report the similarity of the original and the mutated data under two metrics: our selected editing distance and the  embedding similarities  based on the OpenAI text-embedding model:
>
> | Language | Ave. embedding Distance | Ave. editing distance |
> |----------|----------------------------|--------------|
> | Python   | 0.61                       |    0.517    |
> | C/C++    | 0.52                       |   0.379  |
> | Java     | 0.54                       |   0.408  |
>
> Fourth, we conduct experiments comparing three models on secure code generation without security policy on original seeds versus mutated samples.
>
> | Model             | Python    | Python   | C/C++     | C/C++    | Java      | Java     |
> |-------------------|-----------|----------|-----------|----------|-----------|----------|
> |                   | originals | %changed | originals | %changed | originals | %changed |
> | GPT-4o-2024-08-06 | 52%       | +10%     | 18%       | -39%     | 28%       | +18%     |
> | Claude-3.7-Sonnet | 68%       | -4%      | 21%       | +10%     | 46%       | +5%      |
> | DeepSeek-R1-0528  | 55%       | +22%     | 13%       | +31%     | 33%       | -21%     |
>
> We can observe that there are notable differences between seeds and mutated data, where the performance either improves or decreases. Note that here, the goal of mutation is to obtain more data such that our benchmarking evaluation can be statistically more significant. We do not aim to create new data that leads to significant differences in the final result. As such, we do not expect a very large difference in the secure code generation before and after mutation. This may lead to invalid data and data that are no longer related to security.
>
> Note that for Python, the gaps between mutations are relatively small, while C/C++ shows the most significant differences, where mutated samples may exhibit up to 40% lower performance compared to original samples. This is aligned with our statement earlier that C/C++ use fewer mutated data.
>
> We would like to kindly restate that our mutation is a strategy to balance between the human effort and creating a large-scale and high-quality dataset. This is a core challenge for creating code-related benchmarks, especially for domain-specific tasks (like security) which naturally require large human effort. In addition, we are committed to building a live benchmark, where we will continuously collect and create more data to enrich the dataset.
>
> ## 2. Duplicated reference
>
> We thank the reviewer for this reminder, we will fix this in the next version of our paper.

---

> > ### Comment · Reviewer_cur1 · 2025-08-04
> >
> > Thanks for the clarification and results. The new results show that the performance differences are due to the mutated samples, rather than the original seeds alone. This addresses my concerns regarding the potential duplication issue within the benchmark, so I am raising my score to Accept.

---

> > > ### Author Response · Authors · 2025-08-04
> > >
> > > Thank you very much for your thoughtful review and for increasing the score! If you have any further suggestions or feedback, we’d greatly appreciate it.

---

### Official Review · Reviewer_8WCh · 2025-07-02

**Rating:** 5
**Confidence:** 4

**Summary:**

This work introduces a benchmark called SECCODEPLT, which provides a unified suite of 5.9 K executable security samples across four mainstream programming languages (Python, C, C++, and Java), covering 44 actively exploited CWE categories. The benchmark is designed around three core tasks which are Secure Coding, Vulnerability Detection, and Patch Generation. For each sample it supplies a complete artifact chain. The construction pipeline follows a two-stage process: manual seed samples and expansion via LLM, followed by dynamic execution validation. In Stage 2, GPT‑4o is used to rewrite tasks and expand the number of lightweight code mutations, and functional & PoC tests are used to double verify security/functional properties.

In the experiment, three open source models and three closed source models were tested. It was found that SECCODEPLT achieves nearly 100% in both security relevance and instruction faithfulness compared to the previous benchmark, while CYBERSECEVAL only achieves 68%. Among different programming languages, C/C++ is the most difficult for the model, and the reasoning performance of the model with larger parameters is the best. Adding additional security reminders in the experiment can improve the model's ability to find vulnerabilities, but compared with coding ability, LLM's ability in vulnerability detection and patch generation, especially patch generation, still has a lot of space for improvement.

**Additional Feedback:**

Questions:

● Is fixing the Levenshtein threshold at ≤ 0.8 problematic? Why choose 0.8?

● In Figure 4, both Security-relevancy and Faithfulness Judge are executed by LLM. Is it too dependent on LLM? The radar chart is unclear and hard to compare quantitatively, there are no exact values or confidence intervals, so the impact of “with policy” isn’t visually obvious.

● For patch generation, it is not enough to only report the macro success rate. It is recommended to select a few typical failure cases as root-cause: is it caused by semantic understanding, environmental dependence, or compilation chain?

● What is the specific prompt word? How long is it? Have you done prompt engineering adjustment for different languages?

● Just rename / reorder is not enough. Have you considered introducing other transform methods?

● Have you considered other models when rewriting task descriptions with GPT-4o?

● Functions of 5–30 lines represent only a small fraction of industrial code, have you considered scaling up to repository-level mutations?

● Add some patch failure analysis, such as which are the 5 types of CWE with the highest failure rate? What are the reasons behind it?

● It is recommended to give the overlap analysis with the model pre-training corpora in the appendix, complete the prompt, especially the "additional security reminders"

**Dataset Code Accessibility:**

Yes

**Ethical Considerations:**

No, there are no or only very minor ethics concerns

**Final Justification:**

The authors have attempted my concerns and provided sufficient answers.

**Limitations Weaknesses:**

● The mutator relies primarily on renaming and statement reordering, lacking deeper control- or data-flow transformations, generated samples may remain highly homogeneous with their seeds, offering no guarantee of high-quality mutation.

● The paper mentions that this benchmark can help the model pre-training, but the mutation operation of renaming + symmetric statement swap will cause the samples to be highly homogeneous which risks producing overly similar variants that models might “memorize.”

● Different programming languages have different sample collection methods, especially Python, which is handwritten and cannot be well mapped to real CVE.

● Currently, most samples are clipped functions or small scripts, and the application scenarios are limited. Patch Generation is far from the real industrial scenario.

● Arvo/Juliet is a public resource. If the model pre-training has seen similar code, the evaluation results may be overestimated, and additional deduplication or cross-validation is required, so the performance of closed-source models may be overestimated.

● Some poor performance cases in the Results lack detailed analysis.

● Directly using the success rate as the main indicator is too simplistic, and it is difficult to judge how much impact the change has and whether it is accidental.

**Strengths Contributions:**

+ The SECCODEPLT benchmark covers multiple programming languages and scales up significantly compared to prior benchmarks, supporting Secure Coding, Vulnerability Detection, and Patch Generation tasks on the same dataset.

+ Large-scale Data Generation is achieved through the cooperation of manually crafted seeds and LLM, and each seed generates up to 10 variants and distance checks to ensure the diversity of variants.

+ Through functional + security test case verification, it avoids false positives and missing negatives caused by static rules or LLM Judge.

+ Rigorous experiment analysis, considering both closed-source and open-source models, and the results show the effectiveness of SECCODEPLT and the current problems of LLM in patch generation.

+ Readers can directly get the code with vulnerabilities, patches, and test cases, which is convenient for end-to-end training & evaluation, avoiding the additional cost of "static benchmarks" that only provide summary information to downstream research.

---

> ### Author Rebuttal · Authors · 2025-07-30
>
> We thank the reviewer for the insightful feedback. Please see below for our response.
>
> # Main weaknesses
> ## 1. Mutation issues
> First, we would like to clarify that our mutation strategy does not only modify the function names and variable names. Our mutation strategies also include:
> - Loop Transformations: While to for loop; Do-while to while loop; Loop unrolling
> - Conditional Transformations: If-else to switch statements; Boolean expression simplification
> - Method Refactoring: Extract method refactoring; Inline method for simple functions
> - Code Style Changes: From procedural to functional style
>
> Here, we show an example with basic mutations (renaming) and Code Style Changes
> ```java
> // other context...
> public boolean isStringNotEmpty(String myString) throws Throwable {
>   boolean isGreaterThanZero = false;
>   if ((myString != null) & (myString.length() > 0)) {
>     IO.writeLine("The string length is greater than 0");
>     isGreaterThanZero = true;
>   }
>   return isGreaterThanZero;
> }
> ```
>
> ```java
> // other context...
> public boolean validateString(String myString) throws Throwable {
>   return Optional.ofNullable(myString)
>     .filter(s -> s.length() > 0)
>     .map(s -> {
>       try {
>         IO.writeLine("The string length is greater than 0");
>       } catch (Exception e) {}
>       return true;
>     })
>     .orElse(false);
> }
> ```
>
> Second, our mutation is mainly applied to Python, where we generate 9 mutations from one seed. This is because python does not have real-world codebases as seeds and we manually write the seeds. For C/C++ and Java, we obtain enough seeds from the real-world codebase and only generate 1–2 variants per seed. As such, the impact of the mutation is constrained in our benchmark.
>
> Third, we conducted a **similarity filter** to remove too similar data samples. Below, we report the similarity of the original and the mutated data under two metrics: our selected editing distance and the  embedding similarities  based on OpenAI text-embedding model:
>
> |Language|Ave. embedding Distance|Ave. editing distance|
> |-|-|-|
> |Python|0.61|0.517|
> |C/C++|0.52|0.379|
> |Java|0.54|0.408|
> ## 2. Potential Memorization
> Our mutations can mitigate the issue of memorization as it has changed the prompts and code. In addition, the pool performance of the SOTA models show that the benchmark is still challenging. We will make our benchmark as a live one where we continuously include new repos to combat memorization.
>
> ## 3. Python sample collection and real-world mapping
> Sorry for the confusion. **Our Python samples are based on real CVEs**.  While we manually craft the code examples, each is grounded in top severe vulnerabilities as shown in Table 1 in the paper. .
>
> We chose manually creating seeds for Python because there is no existing Python dataset with comprehensive CWE coverage mapped to real-world codebases (similar to Arvo for C/C++ or Juliet for Java).
>
> ## 4. Function-level benchmark
>
> We would like to clarify that our benchmark, especially for C/C++ and Java are repository-level. For vulnerability detection and patch generation, the users need to identify the vulnerable function from the given repo and patch them. Given the poor performance of SOTA models and agents, we provide additional contexts retrieved from the repo. On average, our full context contains:
> - 1,831 tokens (avg across ARVO and Juliet datasets)
> - Information spanning 4.7 files on average (6.1 files specifically in ARVO)
>
> However, users can choose not to use our provided context and extract their own context using their agents.
>
> ## 5. Potential data contamination and memorization
>
> Our mutations can mitigate the issue of memorization as it has changed the prompts and code. In addition, the pool performance of the SOTA models shows that the benchmark is still challenging.  We will make our benchmark as a live one where we continuously include new repos to combat memorization. In addition, we conduct the following analysis to show that our dataset has not been largely contaminated yet.
>
> We use Min-K[1] (top k=20) to detect potential training data contamination. As we do not have access to the logits of commercial models, we use Qwen2.5-32B (released December 2024) as a proxy.
>
> |Dataset Source|Min-K% Prob|
> |-|-|
> |Arvo (Original)|19%|
> |Arvo (Mutated)|11%|
> |Juliet (Original)|57%|
> |Juliet (Mutated)|31%|
>
> The results show that our mutation can largely reduce the memorization concerns. Note that if the score is around or lager than 70%, the corresponding data is deemed as part of the training data [1]. In our case, the scores are all lower than this threshold.
>
> We also evaluate models with knowledge cutoffs before and after August 2024 using our C/C++ data on secure code generation:
>
> |Model|Original|Mutated|
> |-|-|-|
> |GPT-4o-2024-08-06|17%|9%|
> |GPT-4o-2024-11-20|16%|13%|
>
> The results demonstrate no significant performance advantage for models with more recent training data, suggesting that memorization does not substantially influence our evaluation outcomes.
>
> ## 6. Failure case analysis
>
> We provide an example analysis here:
>
> ```cpp
> // information that indicates this assertion
> static constexpr int32 motionOffset[7] = {-4, -2, -2, 0, 0, 2, 4};
> static constexpr int32 motionDoAverage[7] = {0, 0, 1, 0, 1, 0, 0};
>
> int32 slideOffset = motionOffset[motion];
> int32 doAverage = motionDoAverage[motion];
>
> for (uint32 i = 0; i < 16; i++) {
>   ushort16* refpixel;
>
>   if ((row + i) & 0x1) {
>     // Red or blue pixels use same color two lines up
>     refpixel = img_up2 + i + slideOffset;
>
>     if (col == 0 && img_up2 > refpixel)
>       ThrowRDE("Bad motion %u at the beginning of the row", motion);
>
>     // assertion that LLM missed
>     if (col + 16 == width &&
>         ((refpixel >= img_up2 + 16) ||
>          (doAverage && (refpixel + 2 >= img_up2 + 16))))
>       ThrowRDE("Bad motion %u at the end of the row", motion);
>
>   } else {
>     // Green pixel N uses Green pixel N from row above
>     // (top left or top right)
>     refpixel = img_up + i + slideOffset + (((i % 2) != 0) ? -1 : 1);
>
>     if (col == 0 && img_up > refpixel)
>       ThrowRDE("Bad motion %u at the beginning of the row", motion);
>   }
>
>   // In some cases we use as reference interpolation of this pixel and the next
>   if (doAverage)
>     img[i] = (*refpixel + *(refpixel + 2) + 1) >> 1;
>   else
>     img[i] = *refpixel;
> }
>
> img += 16;
> img_up += 16;
> img_up2 += 16;
> ```
>
>
> In this code, the slideOffset comes from `motionOffset[motion]` which can be:
> Positive values: 2, 4
> At the end of the row, positive offsets can push refpixel beyond valid boundaries.
> This can happen in two scenarios:
> (1) `img_up2` points to the start of the reference row (2 rows above)
> img_up2 + 16 points to the end of the current 16-pixel block in the reference row
> refpixel is calculated as: `img_up2 + i + slideOffset`
> If `refpixel >= img_up2 + 16`, it means we're trying to access pixels beyond the current block
> This would be accessing unprocessed or invalid memory locations
> (2) When doAverage is true, the code performs interpolation: `(*refpixel + *(refpixel + 2) + 1) >> 1`
> This requires accessing both `refpixel` and `refpixel + 2`
> If `refpixel + 2 >= img_up2 + 16`, the second pixel for averaging would be outside the valid block
>
> The minimal fix for this issue is:
> ```
> if (
>     ((refpixel >= width) ||
>       (doAverage && (refpixel + 2 >= width))))
> ```
> In models' response, all LLMs miss one of the conditions in the if statement, which is `refpixel + 2 >= img_up2 + 16`.
>
> Besides, we also have some other observations below:
> 1. LLMs struggle with comprehensive analysis of complex code, especially when it involves multiple complex branches.
> 2. LLMs also hallucinate incorrect context, especially in vulnerability detection tasks, where they may reason with critical incorrect information and lead to incorrect conclusions.
>
> ## 7.Metrics beyond success rate
> The unit test rate is the most accurate reflection of whether secure coding and patch generation tasks are completed.
> If phased indicators are needed, the metrics we can consider include BLEU, CodeBLEU, LLM Judge.
> Our evaluation suite is designed with flexibility in mind—researchers can choose from multiple evaluation metrics based on their computational constraints.
> We provide some CodeBLEU results below.
> |Model|Python unit test|Python codebleu|C/C++ unit test|C/C++ codebleu|Java unit test|Java codebleu|
> |-|-|-|-|-|-|-|
> |GPT-4o-2024-08-06|53%|27%|10%|18%|24%|37%|
> |Claude-3.7-Sonnet|68%|24%|21%|17%|46%|30%|
> |DeepSeek-R1-0528|55%|28%|13%|20%|33%|42%|
> ## 8. Levenshtein threshold
> We select the 0.8 threshold based on empirical analysis. Specifically, we manually analyzed 20 samples to determine an appropriate threshold.
>
> ## 9. LLM dependency for evaluation and Figure 4 clarity
> We acknowledge the dependency on LLMs for evaluation. To mitigate potential bias, we conduct experiments on other LLMs including DeepSeek-R1 and Gemini-2.5-pro, beyond the Claude and OpenAI results shown in the paper.
> We achieve an average 0.87 security relevance score, which is much higher than the 0.54 in CyberSecEval.
>
> Regarding Figure 4's clarity, we agree the radar chart could be improved. We will provide a table format with exact values and confidence intervals in the revision.
>
> ## 10. On specific prompts and prompt engineering
> Prompt word: we try different prompt words in Appendix J.1: default prompt, Auto CoT, Manual CoT.
> Average Length: default prompt: 92 tokens, Auto CoT: 101 tokens, Manual CoT: 147 tokens.
> We do not adjust prompts across languages.
>
> ## 11. Using other models for task description rewriting
> We experiment with multiple models for task description generation. Our latest results use O3-mini, and we are continuously updating our dataset.
> The task description is indeed more accurate in human inspection (Claude-3.7-Sonnet achieves 4.2% more in C/C++ Secure Coding), and we plan to continuously update our benchmark with the improved data.
>
> [1] Detecting Pretraining Data from Large Language Models

---

> > ### Comment · Reviewer_8WCh · 2025-08-04
> >
> > Thanks for the rebuttal. It has addressed most concerns. I am happy with it.

---

> > > ### Author Response · Authors · 2025-08-04
> > >
> > > Thank you very much for your thoughtful review! If you have any further suggestions or feedback, we’d greatly appreciate it.

---

### Official Review · Reviewer_hYsW · 2025-07-02

**Rating:** 5
**Confidence:** 3

**Summary:**

SecCodePLT is a multi-lingual, multi-task security benchmark for code LLMs and agents. The tasks include measuring security risks or insecure code generation and security capabilities which includes vulnerability detection and patch generation. The authors claim to cover more CWE categories compared to previous work. They test SOTA models on this benchmark including GPT, Claude, Deepseek, Qwen and cursor agent. Analysis includes interesting language and model specific observations like it is harder for SOTA models to detect vulnerabilities in C, C++ and java code compared to Python.

**Dataset Code Accessibility:**

Yes

**Dataset Code Comments:**

I could access the data and code.

**Ethical Considerations:**

No, there are no or only very minor ethics concerns

**Final Justification:**

The authors provided some clarification for the questions asked. This is a good paper and an Accept according to me.

**Limitations Weaknesses:**

This is a good paper with meaningful contribution. However there are still ways to improve in certain aspects.

1. Evaluation could have been more agent centric. Would have been interesting to see how the results change if you allowed some LLM reflection based on environment feedback.

2. Although the benchmark covers some important languages, other popular languages like JavaScript, typescript, Go and rust are excluded.

3. Although dynamic evaluation is more reliable than static or LLM based evaluation, it is also computationally heavy and may deter some users.

**Strengths Contributions:**

This is a well written paper covering an important topic. Multiple languages are used and analyzed broadening the scope and leading to more useful observations.

1. SOTA models and cursor agent tested, which makes for insightful and relevant baselines.

2. Language specific analysis leads to interesting insights.

3. Dynamic evaluation is more reliable than static rule based or LLM based evaluation.

4. Overall, the scope is reasonably broad with multiple languages, more CWE categories, multiple SOTA models and also an agent.

---

> ### Author Rebuttal · Authors · 2025-07-30
>
> We thank the reviewer for the insightful and positive feedback. Please see below for our response.
>
> **1. Evaluation on agents:**
> Following the reviewer’s comment, we applied OpenHands [1] to our benchmark’s secure coding task. We use the default setting from openhands code repo and change the LLM provider to the following three models:
>
> | Model             | Python        | Python         | C/C++         | C/C++          |
> |-------------------|---------------|----------------|---------------|----------------|
> |                   | Model only | Openhands |Model only | Openhands |
> | GPT-4o-2024-08-06 | 51%           | 54%            | 10%           | 14%            |
> | Claude-3.7-Sonnet | 63%           | 65%            | 19%           | 26%            |
> | DeepSeek-R1-0528  | 53%           | 52%            | 7%            | 8%             |
>
> We observe that agents in general perform better than the models themselves. On python, we observe a smaller improvement mainly contributed by the explicit planning component in the agents. On C/C++, the larger improvement is because of the context retrieval tools (`read_file`,  `scroll_down` and `scroll_up`, etc). They can help extract additional context from the repos.
>
> **2. Language coverage:**
>
> We acknowledge that including JavaScript, TypeScript, Go, and Rust would enhance the benchmark's comprehensiveness. However, adding these languages requires non-trivial effort. In our future work, we will extend our benchmark to broader languages.
> We chose to focus initially on Python, C/C++, and Java as they represent the highest number of security vulnerabilities and cover three major programming paradigms.
>
> **3. Computational cost of dynamic evaluation:**
>
> Dynamic evaluation has been widely used in many coding based benchmarks given their high precision. However, to address the scalability concern, we also support static metrics, including Bleu, CodeBleu[2] and LLM Judge.
> In the following, we show the result of using CodeBleu[2] on the secure coding task, where we compare the similarity between the model generated code and ground truth secure code.
>
> | Model             | Python    | Python   | C/C++     | C/C++    | Java      | Java     |
> |-------------------|-----------|----------|-----------|----------|-----------|----------|
> |                   | unit test | codebleu | unit test | codebleu | unit test | codebleu |
> | GPT-4o-2024-08-06 | 53%       | 27%      | 10%       | 18%      | 24%       | 37%      |
> | Claude-3.7-Sonnet | 68%       | 24%      | 21%       | 17%      | 46%       | 30%      |
> | DeepSeek-R1-0528  | 55%       | 28%      | 13%       | 20%      | 33%       | 42%      |
>
>
> [1]: OpenHands: An Open Platform for {AI} Software Developers as Generalist Agents
>
> [2]: CodeBLEU: a Method for Automatic Evaluation of Code Synthesis

---

> > ### Comment · Reviewer_hYsW · 2025-08-05
> >
> > Good to see agent and static evaluation results.
> >
> > Interesting to see OpenHands helps more with C/C++ then Python. Why do you say it's because of tools and not because of some other reason like base model capability is lacking in C/C++?
> >
> > Agree adding any language is non-trivial effort.
> >
> > Very interesting to see static vs dynamic results. Especially the fact that static results always trail for python, but sometimes exceed dynamic results for other languages.

---

> > > ### Author Response · Authors · 2025-08-05
> > >
> > > Thank you for your feedback! We appreciate your insights and will discuss these points in detail:
> > > ## Tool-calling vs. base model capability in C/C++
> > > We apologize for not being clear in our previous response. When we say this is due to tool calling, it's actually related to both the model's inherent tool-calling capabilities and the tools themselves.
> > > Tool calling brings in more context; once the model has more context, it naturally performs better.
> > > From our experiments, we can observe that Claude 3.7 brings the greatest improvement, followed by GPT-4o, while DeepSeek shows almost no improvement. This can be seen from their respective technical reports - DeepSeek, as a reasoning model, has inherently weak tool-calling performance; GPT-4o-2024-08-06 is relatively older; and Claude 3.7 has the strongest coding and tool-calling capabilities, thus achieving the most improvement.
> > >
> > > Additionally, we agree that this may also reflect the model's weaker base capability in C/C++, as it requires more context to handle these languages effectively.
> > >
> > > ## Static vs. dynamic performance differences across languages
> > >
> > > We think this may be related to the datasets themselves.
> > > It is harder for C/C++ and Java to pass unit tests as they are at the repo level, which is why dynamic testing scores are sometimes lower than static scores.
> > >
> > > Thank you again for raising these points! We will add them to our discussion section in the future version of our paper.

---

### Official Review · Reviewer_vjyB · 2025-07-03

**Rating:** 4
**Confidence:** 4

**Summary:**

This paper introduces a benchmark for Code Language Models that offers (relative to existing benchmarks) higher coverage (in terms of CWEs) and are dynamic i.e. contains artifacts to be able to run the code generated and check vulnerabilities using designated test cases. The benchmark starts with curated seed problems from existing repositories for C/C++, and Java, while for Python authors manually generate entire seed problems entirely (due to lack of existing corpus / repos). The seed data's skeleton contains a task description, root cause (the code chunk leading to the vulnerabilities), and functionality tests. The paper does takes measures to ensure unambiguous specification of desired behavior. The final task description are synthetically generated via GPT-4.1. The benchmark that expands with use Text Description mutation Code mutation i.e. rewritten by GPT4o at high temp. Post mutation, these expanded sets validated and filtered via dynamic evaluation testing correctness.

**Dataset Code Accessibility:**

Yes

**Dataset Code Comments:**

The authors haven't released official repository for eval code of this. Please correct me if that's not the case.

**Ethical Considerations:**

No, there are no or only very minor ethics concerns

**Final Justification:**

Many of my concerns are satisfied from the rebuttal and upon seeing appendix.

**Limitations Weaknesses:**

- For expansion from seed data, the paper used Text + Code mutation Line 253: "To avoid redundancy, we further calculate the new samples’ similarity to seeds using the word-level Levenshtein distance. " In my understanding, the expansion method only copies the problem while preserving the function of each problem. So that is to say, even though benchmark is scale (re: scalable), it is functionally reducible to the starting point. In my understanding, the benchmark ability gains of this expansion is minimal. One way to test it to put up benchmark scores before and after mutation and see what value the expansion provided in isolation. If no improvement, perhaps this expansion doesn't do much. Also, I don't agree with the reasoning given for not using embedding model in line 254: "We choose the editing distance as we observed that it can better capture the instruction-level differences than distances based on embedding models." It appears to me that entire set would reduce to minimal functional set i.e. size of seed problems if you use good enough embeddings for deduplication (which is kinda expected for a benchmark dataset to do provided assertion of being scalable). The experiments cover secure coding experiment and vulnerability for variety of open and closed models.
- Line 283: "Appendix C shows the prompts for both judges and the consistency of judging results with different judgment models" refers to IMO important results in Appendix C. But there is NO appendix (from A to G) in the paper. This is unfortunate and invalidates trusts as a reader / reviewer of this work.
- It would be nice to see impact of model size for same family (like Qwen-2.5-Coder Instruct).

**Strengths Contributions:**

- The introduced benchmark is promising with dynamic evaluation and can end to end benchmark models for given CWEs.
- The paper benchmark variety of model families.

---

> ### Author Rebuttal · Authors · 2025-07-30
>
> We thank the reviewer for the insightful feedback. Please see below for our response.
>
> ## 1. Mutation issues
>
> First, we would like to clarify that our mutation strategy does not only modify the function names and variable names. Our mutation strategies also include:
> - Loop Transformations: While to for loop; Do-while to while loop; Loop unrolling
> - Conditional Transformations: If-else to switch statements; Boolean expression simplification
> - Method Refactoring: Extract method refactoring; Inline method for simple functions
> - Code Style Changes: From procedural to functional style
>
> Here, we show an example with basic mutations (renaming) and Code Style Changes
> ```java
>     // other context...
>     public boolean isStringNotEmpty(String myString) throws Throwable {
>         boolean isGreaterThanZero = false;
>         if ((myString != null) & (myString.length() > 0)) {
>             IO.writeLine("The string length is greater than 0");
>             isGreaterThanZero = true;
>         }
>         return isGreaterThanZero;
>     }
> ```
>
> ```java
>     // other context...
>     public boolean validateString(String myString) throws Throwable {
>         return Optional.ofNullable(myString)
>             .filter(s -> s.length() > 0)
>             .map(s -> {
>                 try {
>                     IO.writeLine("The string length is greater than 0");
>                 } catch (Exception e) {}
>                 return true;
>             })
>             .orElse(false);
>     }
> ```
>
> Second, our mutation is mainly applied to Python, where we generate 9 mutations from one seed. This is because Python does not have real-world codebases as seeds and we manually write the seeds. For C/C++ and Java, we obtain enough seeds from the real-world codebase and only generate 1–2 variants per seed. As such, the impact of the mutation is constrained in our benchmark.
>
> Third, we conducted a **similarity filter** to remove too similar data samples. Below, we report the similarity of the original and the mutated data under two metrics: our selected editing distance and the  embedding similarities  based on OpenAI text-embedding model:
>
> | Language | Ave. embedding Distance | Ave. editing distance |
> |----------|----------------------------|--------------|
> | Python   | 0.61                       |    0.517    |
> | C/C++    | 0.52                       |   0.379  |
> | Java     | 0.54                       |   0.408  |
>
> Four, we conduct experiments comparing three models on secure code generation without security policy on original seeds versus mutated samples.
>
> | Model             | Python    | Python   | C/C++     | C/C++    | Java      | Java     |
> |-------------------|-----------|----------|-----------|----------|-----------|----------|
> |                   | originals | %changed | originals | %changed | originals | %changed |
> | GPT-4o-2024-08-06 | 52%       | +10%     | 18%       | -39%     | 28%       | +18%     |
> | Claude-3.7-Sonnet | 68%       | -4%      | 21%       | +10%     | 46%       | +5%      |
> | DeepSeek-R1-0528  | 55%       | +22%     | 13%       | +31%     | 33%       | -21%     |
>
> We can observe that there are notable differences between seeds and mutated data, where the performance either improves or decreases. Note that here, the goal of mutation is to obtain more data such that our benchmarking evaluation can be statistically more significant. We do not aim to create new data that leads to significant differences in the final result. As such, we do not expect a very large difference in the secure code generation before and after mutation. This may lead to invalid data and data that are no longer related to security.
>
> Note that for Python, the gaps between mutations are relatively small, while C/C++ shows the most significant differences, where mutated samples may exhibit up to 40% lower performance compared to original samples. This is aligned with our statement earlier that C/C++ uses fewer mutated data.
>
> We would like to kindly restate that our mutation is a strategy to balance between the human effort and creating a large-scale and high-quality dataset. This is a core challenge for creating code-related benchmarks, especially for domain-specific tasks (like security), which naturally require large human effort. In addition, we are committed to building a live benchmark, where we will continuously collect and create more data to enrich the dataset.
>
>
> ## 2. Missing appendices
>
> We would like to kindly point out that the appendices (A through G) can be found in  Supplementary Material  (Following NeurIPS guidelines, we separated the main paper from supplementary materials/appendices, which is why they are not visible in the main PDF). We have also made our code and dataset publicly available through the links provided in the submission system.
>
> ## 3. The impact of  model size
>
> Following the reviewer’s suggestion, we conduct the secure coding task with  Qwen-2.5 model family across different sizes
>
> | Model Size            | Secure Coding (Python) | Secure Coding (C/C++) |
> |-----------------------|------------------------|-----------------------|
> | Qwen2.5-1.5B-Instruct | 14%                    | 0.2%                  |
> | Qwen2.5-7B-Instruct   | 22%                    | 0.4%                  |
> | Qwen2.5-14B-Instruct  | 33%                    | 1.3%                  |
> | Qwen2.5-32B-Instruct  | 31%                    | 1.5%                  |
> | Qwen2.5-72B-Instruct  | 49%                    | 4.8%                  |
>
> We observe that, overall, performance is positively correlated with model size.
> We find that for Python, there is a large gap between the 1.5B and 7B models, the 72B model and the 14B/32B models. For C/C++, models below 32B show relatively similar performance, while the 72B model demonstrates a significant improvement.

---

> > ### Author Response · Authors · 2025-08-07
> >
> > Dear reviewer,
> >
> > We would like to kindly follow up on our rebuttal. We are at your disposal to address additional comments. Thank you!

---

> > > ### Comment · Reviewer_vjyB · 2025-08-08
> > >
> > > Thanks for your detailed response and work in general. In light of the review, I have raised the score to accept. :)

---

> > > > ### Author Response · Authors · 2025-08-08
> > > >
> > > > Thank you very much for your thoughtful review and for increasing the score! If you have any further suggestions or feedback, we’d greatly appreciate it.

---

### Note · Authors · 2025-08-12

We thank the reviewers for their constructive feedback and insightful comments.
We have responded to all reviewers and did all the required experiments:

**New experiments**
- **Model size impact analysis**: We evaluated the Qwen-2.5 model family across different sizes (1.5B to 72B), demonstrating positive correlation between model size and performance, with significant improvements at 72B.
- **Agent-based evaluation**: We applied OpenHands to our benchmark, showing that agents generally outperform standalone models, with improvements of 3-7% on Python and 4-19% on C/C++ tasks.
- **Static evaluation metrics**: We implemented CodeBLEU as a static metric, providing a computationally efficient option while maintaining evaluation quality.
- **Contamination analysis**: We conducted Min-K analysis and cross-version model comparisons to demonstrate minimal data contamination in our benchmark.
- **Original vs. mutated performance comparison**: We compared model performance on original seeds versus mutated samples, showing notable differences (up to 40% variance) that validate our mutation approach.
- **Cross-version model evaluation**: We compared GPT-4o models with different knowledge cutoffs (August vs November 2024) on our C/C++ data, showing no significant performance advantage for newer models, further confirming minimal contamination.

Below, we also summarize the key points that don't require experiments:
- We clarified that our mutation strategies extend beyond simple renaming to include loop transformations, conditional transformations, method refactoring, and code style changes. We also provided empirical validation to show the significant differences between original and mutated samples.
- We clarified that our C/C++ and Java benchmarks are repository-level with average contexts spanning 4.7 files and 1,831 tokens.
- We provided detailed failure case analysis demonstrating LLMs' struggles with complex branching logic and comprehensive code analysis.
- We clarified that Python samples are based on real CVEs, manually crafted but grounded in the top severe vulnerabilities from Table 1.
- We justified the 0.8 Levenshtein threshold based on empirical analysis of 20 samples.
- We addressed LLM dependency concerns by testing multiple models (DeepSeek-R1, Gemini-2.5-pro), achieving a 0.87 security relevance score.
- We provided prompt engineering details showing different prompt strategies (default: 92 tokens, Auto CoT: 101 tokens, Manual CoT: 147 tokens).

---

### Decision · Program_Chairs · 2025-09-18

**Decision:**

Accept (poster)

**Comment:**

LLMs are widely used for generating code, but the security of that code is a big concern. Compared to other work in this area, this benchmark is bigger and broader, covering 5.9k samples, 44 CWEs, and three languages (Python, C/C++, Java). It also includes dynamic evaluation, to more rigorously determine if the generated code is actually secure. Overall, this work is quite thorough, and the authors did a good job of addressing reviewer concerns.

===== FINAL UPDATE FROM DB Track PCs ====

The final decision for this paper has been taken by the program chairs after consultation with the SACs. All Senior Area Chairs have ranked papers according to the feedback from the AC during the review process. We decided to leave the original meta-review to reflect the opinion of the AC in light of the initial discussions with reviewers and SAC.